# Confinement Effect and Efficiency of Concentrically Loaded RACFCST Stub Columns

**DOI:** 10.3390/ma15010154

**Published:** 2021-12-26

**Authors:** En Wang, Yicen Liu, Fei Lyu, Faxing Ding, Yunlong Xu

**Affiliations:** 1School of Civil Engineering, Central South University, Changsha 410075, China; wangen718@csu.edu.cn (E.W.); liuyicen796@163.com (Y.L.); dinfaxin@csu.edu.cn (F.D.); xuyunlong@csu.edu.cn (Y.X.); 2Engineering Technology Research Center for Prefabricated Construction Industrialization of Hunan Province, Changsha 410075, China

**Keywords:** recycled aggregate concrete (RAC), concrete-filled steel tube columns (CFST), finite element (FE) analysis, ultimate bearing capacity, confinement effect, constitutive models

## Abstract

Recycled aggregate concrete-filled steel tubular (RACFST) columns are widely recognized as efficient structural members that can reduce the environmental impact of the building industry and improve the mechanical behavior of recycled aggregate concrete (RAC). The objective of this study is to investigate the behavior of recycled aggregate concrete-filled circular steel tubular (RACFCST) stub columns subjected to the axial loading. Three-dimensional finite element (FE) models were established using a triaxial plastic-damage constitutive model of RAC considering the replacement ratio of recycled aggregates. The FE analytical results revealed that the decreased ultimate bearing capacity of RACFCST stub columns compared with conventional concrete infilled steel tubular (CFST) columns was mainly due to the weakened confinement effect and efficiency. This trend will become more apparent with the larger replacement ratio of recycled aggregates. A practical design formula of the ultimate bearing capacity of RACFCST stub columns subjected to axial load was proposed on the basis of the reasonably simplified cross-sectional stress nephogram at the ultimate state. The derivation process incorporated the equilibrium condition and the superposition theory. The proposed equation was evaluated by comparing its accuracy and accessibility to some well-known design formulae proposed by other researchers and some widely used design codes.

## 1. Introduction

The past several decades have witnessed the rapid progress of engineering construction technology and worldwide expedite urbanization, which leads to a continuous updating of design codes to meet the changing requirements of society. Therefore, the Construction & Demolition (C&D) waste is becoming an inevitable problem due to the upgrading of existing engineering structures. As the most commonly used engineering materials by far, concrete has the merits of low price and high compressive strength. According to available survey studies, around 20 billion metric tons concrete were consumed in 2009 [1], and this number exceeded 26 billion metric tons by 2012 [2]. Researchers predicted that the worldwide demand of aggregates will be doubled in the next 20 years [3]. Therefore, social concerns were raised about environmental issues such as carbon dioxide emission, waste of fresh water resources, and energy consumption. Thus, the concrete industry has been seeking a sustainable way to minimize the environmental impact. The recycled aggregate concrete (RAC), as a new type of green building material that emerged in past two decades, can reasonably dispose of and effectively utilize demolished concrete. Hence, RAC is beneficial in terms of saving natural resources and environmental preservation. The waste concrete demolished from existing engineering structures is recycled and crushed as coarse aggregate to produce new concrete. This recycled material has been widely recognized as an attractive and promising alternative to conventional structural concrete among the scientific community.

However, compared with structural concrete utilizing natural aggregates, RAC has some drawbacks verified by experimental studies such as the low strength and elastic modulus, high shrinkage and creep, inferior durability, and large scatter of its tested mechanical properties [4,5,6,7,8]. Those defects stem from the high porosity, the residual adherent mortar, and different sources of waste concrete, which hindered their application in engineering practice. Consequently, the usage of RAC in practical construction usually requires some enhancement treatments regarding the microstructure of recycled aggregates including the heat treatment [9], ultrasonic cleaning [10], carbon dioxide injection [11], etc. [12,13,14]. On the other hand, RAC in some forms of composite structures shows more reliable and uniform mechanical properties and is supposed to overcome the aforementioned application barriers.

One of the most promising structural members in its form that can provide sufficient confinements on concrete and reduce their shrinkage and creep is the concrete-filled steel tubular (CFST) columns [15]. Correspondingly, the recycled aggregate concrete-filled steel tabular (RACFST) columns are a convenient way that can suppress the defects of RAC and improve their mechanical performance. Experimental and numerical studies have already been conducted on the RACFST stub columns with circular and square cross-sections [15,16,17,18,19,20,21,22,23,24]. It can be summarized here that under the axial compression, the structural performance of RACFST stub columns is similar to CFST stub columns in their ultimate bearing capacity and failure modes. Meanwhile, Zhang et al. [20] studied the mechanical performance of RACFST slender columns with circular and square cross-sections under eccentric loading, and they found that the replacement ratio of recycled aggregates has less of an effect on the ultimate bearing capacity compared with the eccentricity and slenderness ratio. Chen et al. [21] used the finite element model (FEM) to investigate the behavior of the RACFST stub columns under axial loading. The validation of the established FE model was verified against the test results, and a parametric study was conducted to propose a design formula for calculating the ultimate axial-bearing capacity based on the modification of the code for the design of a steel–concrete composite structure (DL/T 5085-1999) [22]. Some scholars have found that the existing design codes cannot precisely predict the ultimate axial-bearing capacity of RACFST stub columns and therefore proposed virous formulae [16,21,23,24]. For instance, Wang et al. [15] carried out an experimental study of thirty-nine stub columns to investigate their compressive behavior with normal-strength RACFST stub columns. The results showed that the source and the replacement ratio of recycled aggregates slightly affect the static responses and collapse process of the RACFST specimens. An experimental study that contains 32 RACFST specimens was reported by Lyu [25]. The confinement factor and the material strength were found to be the most influential to the bearing capacity of the columns, while the replacement ratio of coarse aggregate considerably affects the strength index of the RAC. The applicability of using the current design provisions of CFST stub columns to calculate the RACFST columns was evaluated using the experiment data. It was found that the well-known design codes of CFST stub columns more or less underestimate the sectional capacity of RACFST stub columns. It should be noticed here that some of newly proposed formulae are too cumbersome to use in design and lack clear physical meanings despite their higher accuracy.

To sum up, past studies partially addressed the understanding of mechanical behavior of RACFST columns under axial loading and verified its prospect in engineering application. However, more in-depth investigations are required to clearly explain the structural behavior such as the confinement effect and efficiency and their relations with the design parameters. In the meantime, the promotion of RAC usage in engineering practice requires more practical and simple yet accurate design tools. To date, the specific calculation method of the ultimate bearing capacity of RACFST stub columns is still rare, and the current design formulae of CFST stub columns were reported as not appropriate. The differences between the confinement effect of the steel tube and infilled RAC and that of normal concrete counterparts should be noticed, including the discrepancy of stress variation of infilled RAC and normal infilled concrete and the correspond steel tube stress variation.

To this end, on the basis of past research of CFST stub columns under axial loading [26], this study investigated the confinement effect and efficiency of RACFCST stub columns and proposed a practical design formula with concise physical sense for the calculation of their axial-bearing capacity. The main contents of this paper include the following. (1) A full-scale FE model of RACFST stub columns was established by adopting the triaxial plastic damage constitutive model of recycled aggregate concrete considering the replacement ratio of recycled aggregate. (2) Available test results of axially loaded RACFCST stub columns were collected. Non-linear FE analysis was conducted to reproduce the collected test results. (3) A parametric study was performed on the basis of the validated FE model to investigate the different confinement effect and efficiency between CFCST and RACFCST stub columns. (4) By means of the superposition method and a large number of numerical results, a practical design formula with concise physical sense was derived. (5) The performance of the proposed design formula was compared with the formulae of well-known international codes.

## 2. Finite Element Modeling of RACFCST Stub Columns

### 2.1. Modeling Method

ABAQUS is widely regarded as the most powerful finite element software for the analysis of complex solid mechanical structural mechanical systems, especially for the control of very large and complex problems and the simulation of highly non-linear problems.

Non-linear FE models of axially compressed RACFCST stub columns were established making use of the commercially available software ABAQUS version 6.14 [27]. The outer steel tube, infilled RAC, and the loading instrument were modeled by the eight-node reduced integral format 3D solid element (C3D8R) provided by the software element library.

The mesh size will affect the accuracy of the FE model and computation time. Choosing a proper mesh size is important to reasonably replicate the test results. In order to identify the optimal mesh size, a convergence study was performed. The scope of the mesh size is from *D*/25 to *D*/5, where *D* means the outer diameter of RACFCST stub columns. Figure 1 presents the influence of mesh size on the ultimate bearing capacity. The results revealed that the model with *D*/10 mesh size and those with smaller mesh size predicted similar axial-bearing capacity with the difference being less than 1%. The results predicted by the larger mesh size are sensitive to the change of element size. Therefore, the mesh size is selected as *D*/10 in the modeling. The mesh generation and boundary conditions of the established FE model are shown in Figure 2.

The interaction between the steel tube and the infilled RAC was conducted in a surface-to-surface contact model. The main surface is the inner surface of steel pipe, and the secondary surface is the outer surface of concrete. The sliding formulation was chosen as “finite sliding”, and the discretization method was chosen as “surface-to-surface”. The contact properties between the steel pipe and infilled RAC were simulated by normal and tangential behavior. In the normal behavior setting, the “Pressure-Overclosure” was set to “hard” contact mode, and the separation after contact was allowed. The penalty function was set as the tangential behavior of the contact attribute, in which the friction coefficient was 0.5 [7]. A tie constraint may couple two separate surfaces so that no relative motion occurs between them. Thus, in order to ensure that the axial load is applied to the whole column section at the same time in the loading process, the “tie” constraint mode provided by ABAQUS was adopted to constrain the loading plate (main surface) and the RACFCST stub column (secondary surface). The loading plate was assumed to be rigid and was simulated by approximate rigid material in which the elastic modulus was taken as 1.0 × 10^12^ MPa and the Poisson’s ratio was set as 1.0 × 10^−7^.

### 2.2. Constitutive Relation of Infilled Recycled Aggregate Concrete

In this study, the uniaxial full-range stress–strain relation of RAC proposed by Ding et al. [28] was applied to express the compressive behavior of the infilled RAC. This model modified the previous proposed stress–strain relation of conventional concrete [29] and has been verified to be valid in past investigations [30]. The full range stress–strain relation of RAC can be represented by Equation (1).
(1)$$y=\left\{ \begin{array}{ll} \frac{A_{1(r)}x+(B_{1(r)}-1)x^{2}}{1+(A_{1(r)}-2)x+B_{1(r)}x^{2}} & x\leq 1\\ \frac{x}{\alpha_{1(r)}(x-1{)}^{2}+x} & x>1 \end{array}\right.$$
(2)$$\begin{array}{l} A_{1(r)}=\frac{\left(1-0.3r)(1+0.2r\right)}{\left(1-0.1r\right)}\times 9.1f_{\mathrm{cu}}{}^{-4/9}\\ B_{1(r)}=5{\left(A_{1(r)}-1\right)}^{2}/3\\ y=\sigma /f_{c(r)},\;x=\varepsilon /\varepsilon_{c(r)}\\ \alpha_{1(r)}=2.5\times 10^{-5}f_{\mathrm{cu}}{}^{3}\\ f_{c\left(r\right)}=\left(1-0.1r\right)f_{c},f_{\mathrm{cu}\left(r\right)}=\left(1-0.1r\right)f_{\mathrm{cu}}\\ \varepsilon_{c\left(r\right)}=\left(1+0.2r\right)\varepsilon_{c},E_{c\left(r\right)}=\left(1-0.3r\right)E_{c} \end{array}$$
where *r*, *A*_1(*r*)_, *B*_1(*r*)_, *α*_1(*r*)_, *f*_c(*r*)_, *f*_c_, *f*_cu(*r*)_, *f*_cu_, *ε*_c_, *ε*_c(*r*)_, *E*_c_, and *E*_c(_*_r_*_)_ are the parameters to determine the uniaxial behaviors of the RAC; their detailed definition can be found in the reference [29]. The *σ* and *ε* are the stress and strain of RAC, respectively. 

Combining the *σ-ε* relation of RAC under uniaxial compression and the strength criterion parameters of concrete under multiaxial stress [29], the triaxial constitutive model of RAC was derived. The plastic damage model in ABAQUS was adopted, and its parameters were defined as shown in Table 1 [26,29].

The diagrams of confined and unconfined plastic-damage model of recycled aggregate concrete are shown in Figure 3.

### 2.3. Constitutive Relation of Steel Tube

The adopted stress–strain model of an outer steel tube is an elasto-plastic model. Meanwhile, this model considers the Von Mises yield criteria, Prandtl–Reuss flow rule, and isotropic strain hardening. The expression is as follows [29]:(3)$$\sigma_{i}=\left\{ \begin{array}{ll} E_{s}\varepsilon_{i} & \varepsilon_{i}\leq \varepsilon_{y}\\ f_{s} & \varepsilon_{y}<\varepsilon_{i}\leq \varepsilon_{st}\\ f_{s}+\zeta E_{s}(\varepsilon_{i}-\varepsilon_{st})\; & \varepsilon_{\mathrm{st}}<\varepsilon_{i}\leq \varepsilon_{u}\\ \;f_{u} & \varepsilon_{i}>\varepsilon_{u} \end{array}\right.$$
where *σ_i_* is the equivalent stress; *f*_u_ is the ultimate strength of steel, which is equal to 1.5 times the yield strength *f*_s_; the Young’s modulus *E*_s_ is 2.06 × 10^5^ MPa; *ε_i_* stands for the equivalent strain of steel; *ε_y_* means the yield strain; *ε_st_* means the hardening strain and is equal to 12 times *ε_y_*; *ε*_u_ is the ultimate strain and is equal to 120 times *ε_y_*; and *ζ* is a parameter chosen as 1/216. This steel model has been verified in previous studies of concrete-filled steel tubular columns subjected to axial loading [31,32,33,34], shear [35], torsion [36], and cyclic loading [37]. The stress–strain relations of steel derived by the adopted model with different yield strengths is shown in Figure 4.

### 2.4. Model Validation

The validity of the FE models was verified by comparing the existing RACFCST axial compression stub column test data [38,39,40,41,42,43,44,45,46] with the FE calculation results. Since the test method of mechanical properties of concrete and the compressive strength indexes adopted by various scholars are different, the concrete strength conversion formulae (cylinder strength $f_{c}^{\prime }$ to concrete cube strength *f*_cu_ [47], cube with side length of 100 mm *f*_cu,100_ to 150 mm *f*_cu,150_ [29]) are shown in Equations (4) and (5). In this study, the unified compressive cubic strength of 150 mm cube was adopted, which is represented by *f*_cu_^r^_150_ in Table 2.
(4)$$f_{c}^{\prime }=\left\{ \begin{array}{ll} 0.8f_{\mathrm{cu}} & f_{\mathrm{cu}}\leq 50\mathrm{MPa}\\ f_{\mathrm{cu}}-10 & f_{\mathrm{cu}}>50\mathrm{MPa} \end{array}\right.$$
(5)$$f_{\mathrm{cu},150}=1.17\;f_{\mathrm{cu},100}^{0.95}-0.7$$

To verify the applicability and accuracy of the established FE model, the calculated ultimate bearing capacity of collected specimens was compared with the test results. The ratios of the test results to FE results (*N*_u,exp_/*N*_u,FE_) are presented in Table 2. According to the normalized comparison, the mean value of the ratios (*N*_u,exp_/*N*_u,FE_) was 0.978, and the variable coefficient is 0.048, which indicates that the ultimate bearing capacity predicted by the established FE model is in good agreement with the experimental results. Meanwhile, Figure 5 compares some typical load–strain curves obtained by FE analysis and the experiment. It can be seen that the ultimate bearing capacity and initial stiffness obtained from FE analysis fit well with the test results. It is worth mentioning that the FE curves before the peak load point match well with the test curves. Due to concrete crushing or local buckling of steel tubes, the test curves are slightly lower than the FE curves at the later loading stage. The comparisons indicate that the FE model is in good agreement with the experimental results, especially in the prediction of ultimate bearing capacity. Hence, the FE models in this paper can be used for the further numerical study of the RACFCST stub columns under axial compression. Additionally, it is worth mentioning that the average axial strains of the specimen C1, C2, and C3 shown in Figure 5b are converted from the full-height compression of the specimens obtained from the displacement gauge, which could result in a smaller elastic stiffness.

## 3. Investigation of Confinement Effect and Efficiency of RACFCST Stub Columns under Axial Loading

### 3.1. Parametric Analysis

To investigate the axial loaded behaviors of RACFCST stub columns, a parametric study including 75 full-scale FE models based on the aforementioned validated FE modelling approach is conducted. The following are the main parameters analyzed in this study, including size and strength: the diameter of composite section *D* = 500 mm; the columns height *L* = 1500 mm; the tube wall-thickness *t* = 3 mm, 6 mm, and 10 mm, respectively; the steel ratios *ρ* range from 2 to 8%; the strength matching of concrete and steel and the replacement rate of recycled aggregate are shown in Table 3. *f*_cu,0_ means the plain concrete cubic compressive strength. The influence of *f*_cu,0_, *f*_s_, *ρ*, and *r* on the axial loaded behaviors is presented in Figure 6. It can be found that semblable to the CFCST stub columns, the material strengths, and specimen dimensions have a significant effect on the ultimate bearing capacity of RACFCST stub columns.

### 3.2. Evaluation Criterion of the Confinement Effect and Efficiency

The confinement effect is the most valued characteristic of CFST columns caused by the interaction between the steel tube and infilled concrete. The confinement effect can be evaluated by the radial stress *σ_r,_*_c_ of infilled concrete at the ultimate state. In general, the infilled concrete will show better mechanical performance under the stronger confinement effect provided by the outer steel tube. However, the radial stress at the ultimate state cannot reflect the utilization efficiency of the steel tube strength and the variation of composite action during the loading process. Hence, two indexes were investigated in this study to evaluate the appropriateness of different design parameters. The utilization efficiency of steel strength can be evaluated by the radial concrete confinement coefficient (*ξ**_r,c_* = *σ**_r,c_*/(*ρ**f*_s_)). The composite action can be assessed by the occurrence of the intersection point of the axial stress–strain curve and the transverse stress–strain curve of a steel tube. In general, the greater the radial concrete confinement coefficient is, the better the utilization efficiency of the steel tube on the infilled concrete. The composite action can be reflected by the axial and transverse stress–strain curves of the steel, and the intersection of axial and lateral stress–strain curves indicate the best composite action point of the composite structures. The two stress–strain curves intersecting early means the composite action and the confinement efficiency were better [48]. Hence, the aforementioned indexes were utilized herein to assess the confinement effect and efficiency as the design parameters change.

#### 3.2.1. Concrete Strength

The influence on the confinement effect and efficiency of the RACFCST stub columns caused by different concrete strength (*f*_cu_) is shown in Figure 7. Figure 7a,b demonstrate that as the loading progress goes on, the radial stress *σ**_r_*_,c_ and the radial concrete confinement coefficient *ξ**_r,c_* of *f*_cu,0_ = 70 MPa are slightly lower in the early stage but are greater in the late stage. Meanwhile, the longitudinal and the lateral stress–strain curve of *f*_cu,0_ = 70 MPa intersected earlier. It can be concluded that the effect and efficiency of confinement increase with the increase of concrete strength.

#### 3.2.2. Steel Strength

The influence on the confinement effect and efficiency of the RACFCST stub columns caused by diverse steel strength (*f*_s_) is shown in Figure 8. The radial concrete confinement coefficient *ξ**_r,c_* of *f*_s_ = 235 MPa is higher than that of *f*_s_ = 345 MPa in Figure 8a, and the longitudinal and the lateral stress–strain curve of *f*_s_ = 235 MPa intersect before those of *f*_s_ = 345 MPa in Figure 8b. Thus, the high strength steel causes a low efficiency of confinement on the infilled concrete.

#### 3.2.3. Steel Ratio

The influence on the confinement effect and efficiency of the RACFCST stub columns caused by various steel ratios (*ρ*) is shown in Figure 9. From Figure 9a,b, the increase of steel ratio attenuates the radial concrete confinement coefficient *ξ**_r,c_* and postpones the cross of the longitudinal and the lateral stress–strain curves, which means the decrease of confinement efficiency. In conclusion, a thicker steel tube causes a weaker confinement efficiency on the infilled concrete.

#### 3.2.4. The Replacement Ratio of Recycled Aggregate

Figure 7, Figure 8 and Figure 9 present the influence of a diverse replacement ratio of recycled aggregate (*r*) on the confinement effect correlation curves. The comparison states follow the mechanical behavior of RACFCST stub columns:(1)The increase of replacement ratio of recycled aggregate *r* results in the radial concrete confinement coefficient *ξ_r,c_* of RACFCST stub columns being smaller at the initial phase of the loading process. However, at the end of loading, *ξ_r,c_* with different *r* tends to be the same.(2)The axial stress decreased after reaching the peak point, the transverse stress gradually increased under continuous axial loading, and the axial stress–strain curve intersects with the transverse stress curve in the end. However, the intersection point of the longitudinal and the lateral stress–strain curve is slightly postponed with the increase of the recycled aggregate replacement ratio.

The above observations indicate that the replacement ratio of recycled aggregate *r* has less of an effect on the confinement effect and efficiency contrasted with material strengths and steel ratio investigated in this paper.

### 3.3. Contrast of CFCST and RACFCST Stub Columns under Axial Compression

After fixing parameters (*D* = 500 mm, *L* = 1500 mm, *t* = 6 mm, *f*_s_ = 345 MPa, *f*_cu,0_ = 50 MPa) and changing the replacement ratio of recycled aggregate *r*, the concrete stress nephogram values at the middle height cross-section of CFCST and RACFCST stub columns were obtained by ABAQUS calculation, as shown in Figure 10. The increase replacement ratio of recycled aggregate *r* slightly weakens the confinement effect of the steel tube, but the reduction is not obvious.

In order to mathematically analyze the constraint effect of concrete-filled steel tubular columns, a simplified diagram (Figure 11) is used in this section to better understand the derivation of theoretical equations. The simplified model abides by the stress distribution and the superposition theory when the infilled concrete reached the ultimate limit state where *A*_c_ and *A*_s_ are the cross-sectional areas of the concrete and steel tube, respectively. *D* and *D*_0_ are the diameter of the steel tube and infilled concrete, respectively. The areas of the concrete and steel tube could be expressed as:(6)$$\left\{ \begin{array}{l} A_{c}=\frac{\pi D_{0}^{2}}{4}\\ A_{s}=\frac{\pi (D^{2}-D_{0}^{2})}{4} \end{array}\right..$$

## 4. Practical Design Formula of Load-Bearing Capacity of RACFCST Stub Columns

### 4.1. Formulation

According to Figure 11, the relationship between *σ_r_*_,c_ and *σ_θ_*_,s_ in the inelastic stage can be expressed as [48]:(7)$$\sigma_{r,c}=\frac{\rho }{2(1-\rho )}\sigma_{\theta ,s}$$
(8)$$\rho =\frac{D^{2}-D_{0}^{2}}{D^{2}}.$$

For the RACFCST stub columns, the following expressions were derived. *σ_L_*_,s_ and *σ_θ_*_,s_ represents the longitudinal stress and lateral stress of the steel tube, respectively. *σ_L_*_,c_ and *σ_r,_*_c_ represents the axial compressive stress and radial stress of core concrete, respectively.

As shown in Figure 12a,b, when RACFCST stub columns reach the ultimate strength (*f*_sc_ = *N*_u_/*A*_sc_, *A*_sc_ = *A*_c_ + *A*_s_), the average ratio of *σ_L_*_,s_ and *σ_θ_*_,s_ of the steel tube could be obtained as:(9)$$\sigma_{L,s}=0.67f_{s}$$
(10)$$\sigma_{\theta ,s}=0.55f_{s}.$$

The longitudinal stress of confined concrete could be given as:

(11)$$\sigma_{L,c}=f_{c}+k\sigma_{r,c}$$
where *k* is the coefficient of the lateral pressure and *k* = 3.4 was suggested herein, according to Ding et al. [48].

Based on the static equilibrium method, the ultimate bearing capacity *N*_u_ of RACFCST stub columns under axial compression could be put as:(12)$$N_{u}=\sigma_{L,c}A_{c}+\sigma_{L,s}A_{s}.$$

Substituting Equations (7)–(12) into Equation (13), *N*_u_ could be obtained as:(13)$$N_{u}=f_{c}A_{c}+Kf_{s}A_{s}$$
where *K* is the proposed confinement coefficient of RACFCST stub columns subjected to the axial load. Based on the aforementioned FE model analysis, the *K* of the RACFCST stub columns obtained from Equation (13) is 1.61. It worth mentioning that Ding et al. [48] established the equation for the ultimate bearing capacity (*N*_u_ = *f*_c_*A*_c_ + 1.7 *f*_s_*A*_s_) of the CFCST stub column according to the limit equilibrium method, which was validated by 115 CFCST stub columns test results. Meanwhile, Ding et al. [49] combined the FE analysis and the superposition theory to obtain the confinement coefficient *K* (*K* = 1.62) of CFCST stub columns as well. This discrepancy between the two methods was considered due to the constitutive relation of the steel tube in the FE analysis, which could not perfectly reflect the stress–strain curves of the steel tube in the test, especially in the hardening stage. According to the existing experimental tests, local buckling could be occasionally observed on the outer steel tube, while the specimen reached the ultimate loading capacity [31], and the hardening of steel enhances the confinement effect of the steel tube on the infilled concrete. Hence, compensating for the confinement effect of the steel tube in FE analysis and combining with the limit equilibrium method and experimental data verification, the confinement coefficient of CFCST stub columns subjected to axial load is recommended as 1.7. The confinement coefficient *K* of the RACFCST stub columns obtained in this study is close to the confinement coefficient *K* (*K* = 1.62) of CFCST stub columns [49] obtained by FE model analysis, and the radial stress–axial strain curves shown in Figure 7, Figure 8 and Figure 9 indicate that the replacement ratio of recycled aggregate has a minor impact on the confinement effect of the steel tube on the infilled RAC at the ultimate state; thereby, the confinement coefficient of RACFCST and CFCST stub columns is unified as 1.7 here for the sake of precision and simplicity.

### 4.2. Formula Validation

Figure 13 and Table 2 compare the ultimate bearing capacity calculated obtained from Equation (13) (*N*_u_), test results (*N*_u,exp_), and FE results (*N*_u,FE_) for RACFCST stub columns under axial loading. For comparison, the average stress of the cross-sectional area of the column specimens (*N*/(*A*_s_ + *A*_c_)) is used as the horizontal axis. Figure 13a–c show that the average cross-sectional stress calculated by the proposed equation match well with that of *N*_u,exp_ and *N*_u,FE_. The mean value of the ratios of *N*_u,exp_/*N*_u,FE_ is 0.978 with the corresponding dispersion coefficient of 0.048. The average ratio of *N*_u,exp_ to *N*_u_ is 1.024 with the corresponding dispersion coefficient of 0.042. The average ratio of *N*_u,FE_ to *N*_u_ is 1.049 with the corresponding dispersion coefficient of 0.041. As a result, the ultimate axial bearing capacity of RACFCST stub columns calculated by Equation (13) is accurate.

### 4.3. Comparison of Existing Formulae

In order to visually illustrate the advantages of the proposed equation, it is compared with some current international design codes and well-known design formulae proposed by other scholars. Firstly, as shown in Figure 14 and Table 4 and Table 5, the performance and applicability of the current design codes were investigated. Considering the applicable scope of each design code, as shown in Table 4, the number of calculation examples involved in the calculation is different, as shown in the column “Total” in Table 5. Figure 14 presents the predicted axial-bearing capacity using various design codes and indicates that most of the current codes considerably underestimate the bearing capacity. Meanwhile, the average ratios of *N*_u,exp_ to *N*_u,ACI-318-11_, *N*_u,T/CECS 625-2019_, *N*_u,EC4_, *N*_u,AIJ_, and *N*_u,AISC 360-16_ shown in Table 5 are 1.429, 1.188, 1.103, 1.267, and 1.352 with the corresponding dispersion coefficients of 0.060, 0.046, 0.037, 0.037, and 0.069, respectively. Therefore, the proposed formula (Equation (13)) is more precise than the current design codes and has a wider range of applications.

Then, the performance of the proposed design equation was further evaluated by comparing the accuracy of the predicted results using design formulae proposed by other scholars. The details of those formulae are show in Table 6. The predicted results are shown in Figure 15. The average ratios of *N*_u,exp_ to *N*_u,Chen, Z.P.,_ *N*_u,Chen, M.C._, and *N*_u,Xu, W._ are 1.035, 1.268, and 1.025 with the corresponding dispersion coefficients of 0.072, 0.064, and 0.079, respectively. As the result, only the formulae proposed by Xu et al. [24] show a comparable accuracy with the proposed formula (Equation (13)). However, as shown in Table 4, the discreteness of predicted results is relatively larger, and the formulae are more complicated and cumbersome. In general, compared with other scholars’ formulae and international codes, the proposed calculation equation (Equation (13)) is more precise and compendious. Meanwhile, the “confinement effect” is quantified in the derivation of the equation, which explains the influence of the confinement effect on the ultimate bearing capacity more directly.

## 5. Conclusions

In this paper, numerical study was conducted using the FE models validated by existing experimental results to investigate the confinement effect and confinement efficiency of RACFCST stub columns under axial loading. A simple and practical design equation is proposed to predict the ultimate bearing capacity of RACFCST stub columns. The following conclusions could be stated:The FE models of RACFCST stub columns under axial compression can be modeled utilizing the triaxial RAC plastic-damage and steel constitutive relations. The good agreement between the experimental results and the FE calculation results indicates the validity of the FE model.On the basis of verifying the FE modeling method, parameter analysis was conducted on the full-scale RACFCST stub columns. The analytical results indicate that the recycled aggregate slightly weakened the confinement effect between the steel tube and infilled RAC.The conclusions can be drawn from parametric analysis, the *ξ**_r,c_*–axial strain curves and steel stress–axial strain curves indicate that the materials’ strength (*f*_cu_, *f*_s_) and specimen dimensions (*ρ*) impact on the confinement efficiency in the axial loading process. However, the effect of replacement rate of recycled aggregate (*r*) on the confinement effect and efficiency of RACFCST in the limit state can be ignored.An equation for calculating the ultimate bearing capacity of RACFCST stub columns was proposed by the superposition method. The confinement coefficient (*K*) of RACFCST axial compression stub columns is 1.7. The proposed equation is more precise and compendious than current international design codes and other scholars’ literature.

There are still some limitations in this paper. The influence of different kinds of recycled aggregates on the bearing capacity of the RACFCST columns and the application of recycled concrete in practical engineering will be carried out in the future.

## Figures and Tables

**Figure 1 materials-15-00154-f001:**
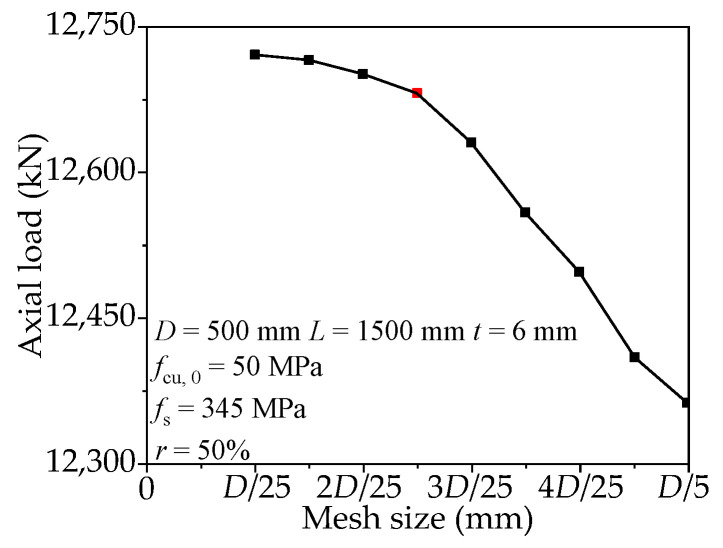
Axial load–mesh size curve.

**Figure 2 materials-15-00154-f002:**
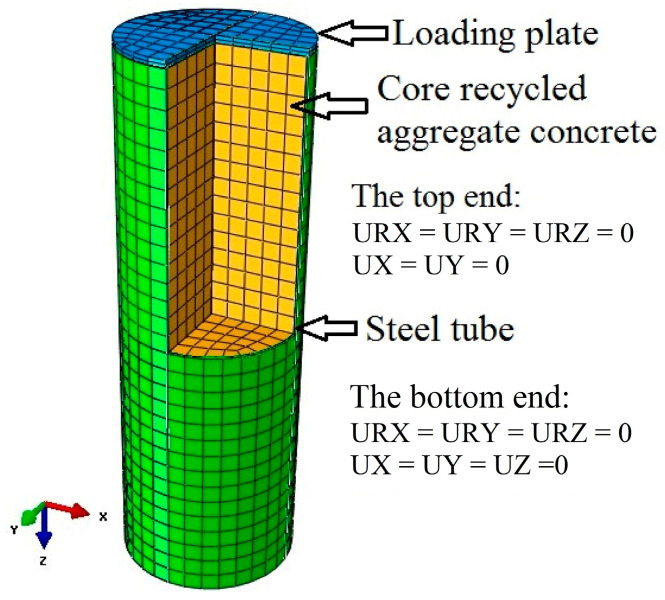
Meshing and boundary conditions of the RACFCST stub column FE model.

**Figure 3 materials-15-00154-f003:**
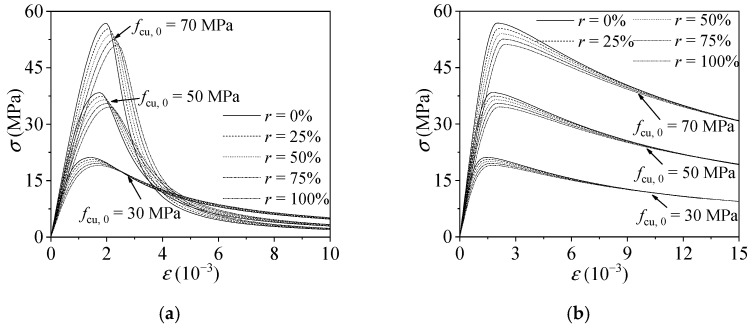
Diagram of confined and unconfined plastic-damage model of RAC: (**a**) unconfined strain–stress relation of RAC with different replacement ratio; (**b**) confined stress–strain relation of RAC with different replacement ratio.

**Figure 4 materials-15-00154-f004:**
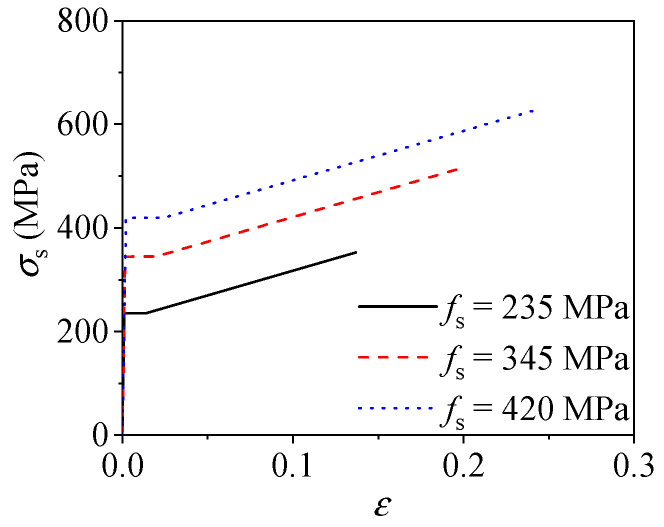
Constitutive relations of steel tube with different yield strength.

**Figure 5 materials-15-00154-f005:**
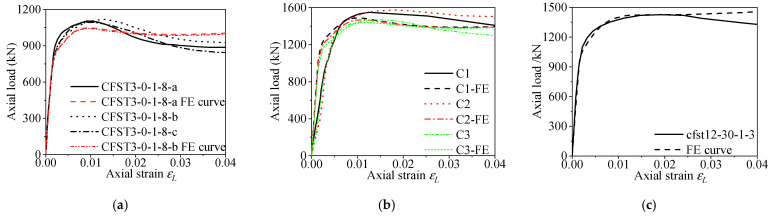
Comparison of FE and experimental load–strain curves: (**a**) reference [38]; (**b**) reference [41]; (**c**) reference [44].

**Figure 6 materials-15-00154-f006:**
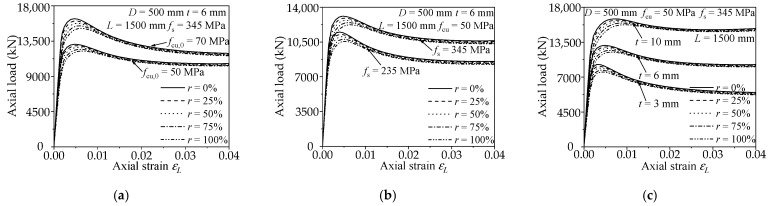
Influence of various parameters: (**a**) concrete strength; (**b**) steel strength; (**c**) steel ratio.

**Figure 7 materials-15-00154-f007:**
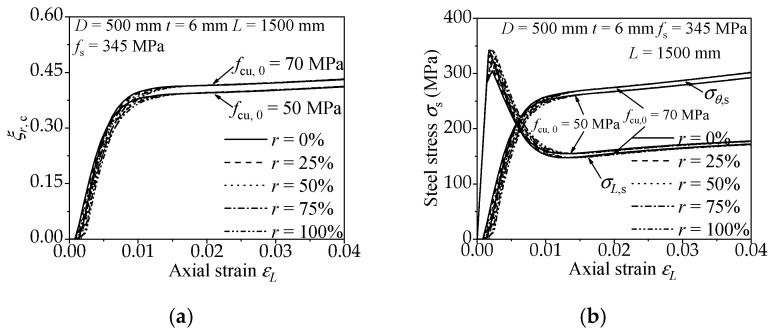
Effect of concrete strength on confinement effect of RACFCST stub columns: (**a**) radial stress–axial strain curves; (**b**) *ξ**_r,c_*–axial strain curves.

**Figure 8 materials-15-00154-f008:**
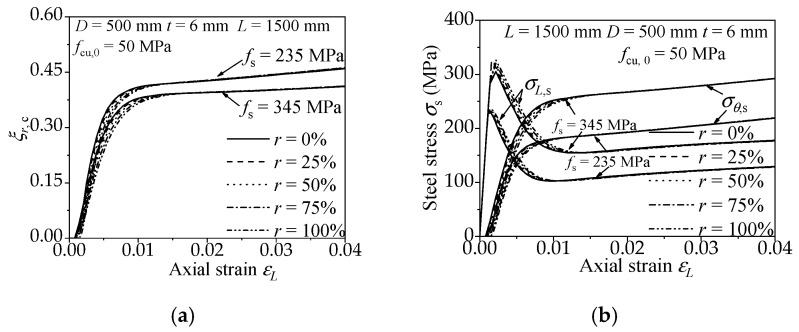
Effect of steel strength on confinement effect of RACFCST stub columns: (**a**) radial stress–axial strain curves; (**b**) *ξ**_r,c_*–axial strain curves.

**Figure 9 materials-15-00154-f009:**
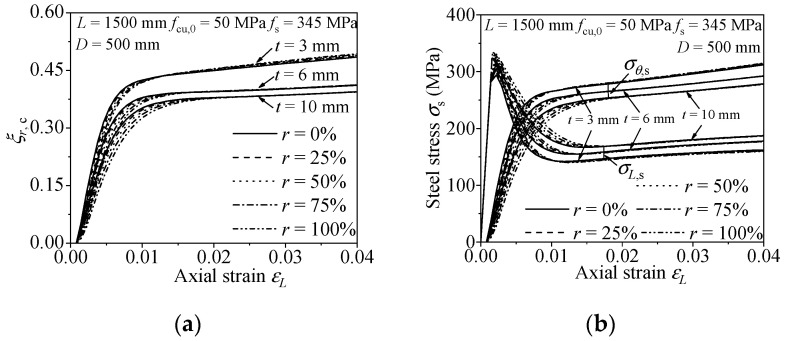
Effect of steel ratio on the confinement effect of RACFCST stub columns: (**a**) radial stress–axial strain curves; (**b**) *ξ**_r,c_*–axial strain curves.

**Figure 10 materials-15-00154-f010:**
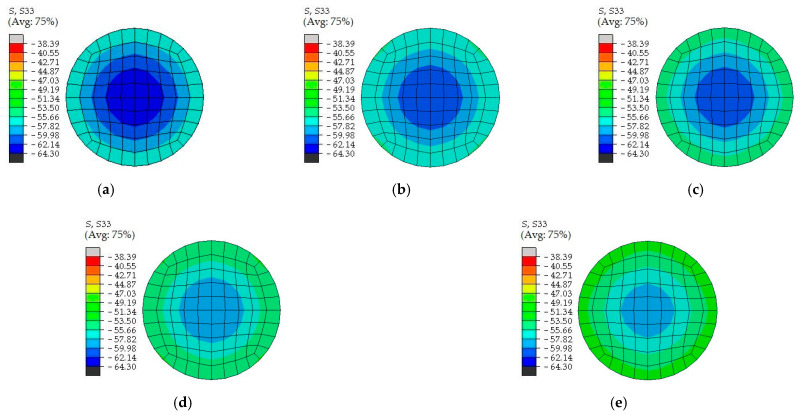
Comparison of stress nephogram of infilled concrete at the mid-section for CFCST and RACFCST stub columns: (**a**) *r* = 0%; (**b**) *r* = 25%; (**c**) *r* = 50%; (**d**) *r* = 75%; (**e**) *r* = 100%.

**Figure 11 materials-15-00154-f011:**
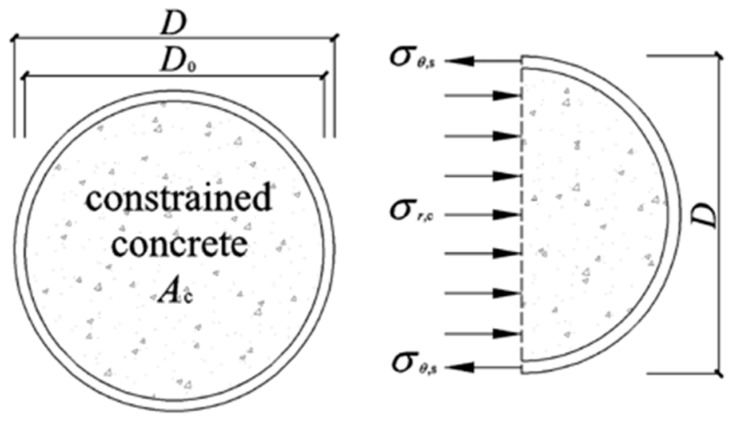
Simplified stress distribution model at the mid-height section of CFCST stub columns.

**Figure 12 materials-15-00154-f012:**
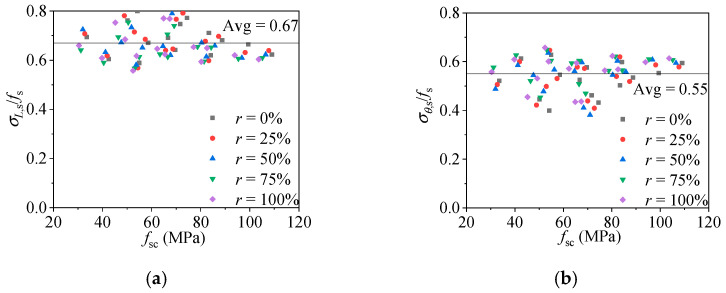
Axial compressive stress and transverse stress of RACFST stub column: (**a**) average ratio of axial compressive stress to the yield stress of the steel tube; (**b**) average ratio of tensile transverse stress to the yield stress of the steel tube.

**Figure 13 materials-15-00154-f013:**
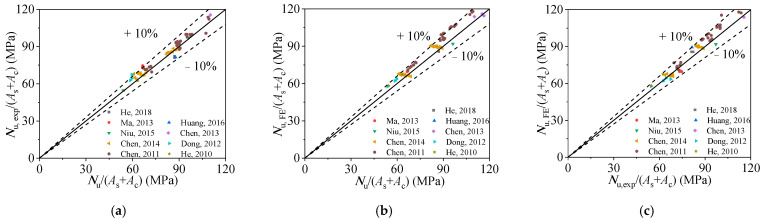
Comparisons of experimental results and FE results with Equation (13): (**a**) comparison of ultimate bearing stress obtained from test results and Equation (13); (**b**) comparison of ultimate bearing stress obtained from FE results and Equation (13); (**c**) comparison of ultimate bearing stress obtained from FE results and test results.

**Figure 14 materials-15-00154-f014:**
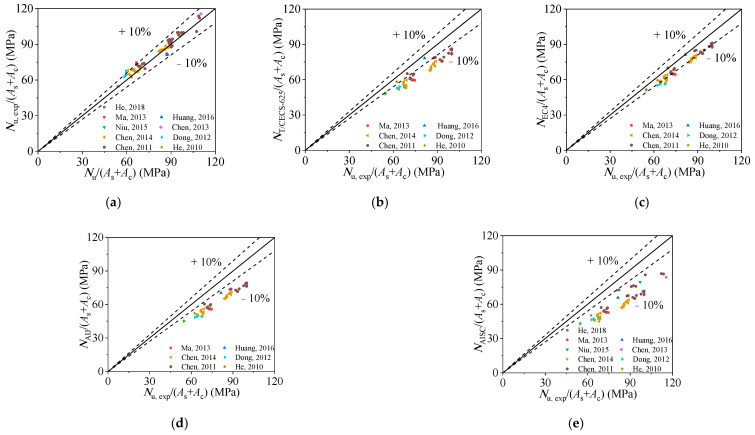
Comparison of experimental results and design methods’ results: (**a**) comparison of ultimate bearing stress obtained from test results and ACI-318-11 [50]; (**b**) comparison of ultimate bearing stress obtained from test results and T/CECS 625-2019 [51]; (**c**) comparison of ultimate bearing stress obtained from test results and EC4 [52]; (**d**) comparison of ultimate bearing stress obtained from test results and AIJ [53]; (**e**) comparison of ultimate bearing stress obtained from test results and AISC 360-16 [54].

**Figure 15 materials-15-00154-f015:**
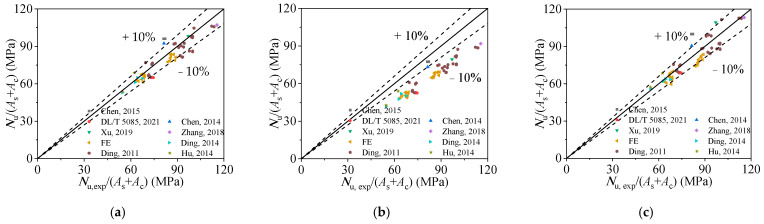
Comparison of experimental results and design methods’ results: (**a**) comparison of ultimate bearing stress obtained from test results and Reference [16]; (**b**) comparison of ultimate bearing stress obtained from test results and Reference [23]; (**c**) comparison of ultimate bearing stress obtained from test results and Reference [24].

**Table 1 materials-15-00154-t001:** Plastic damage parameters of the triaxial constitutive model of RAC.

Dilation Angle	Eccentricity	*f*_b0_/*f*_c0_	K	Viscosity Parameter
40	0.1	1.225	0.6667	0.005

**Table 2 materials-15-00154-t002:** Comparison of loading bearing capacity obtained from experimental and predicted results.

Specimens	Ref.	*D* × *t* × *L* (mm)	*r*	*f*_cu_^r^_150_ (MPa)	*f*_s_ (MPa)	*N*_u,exp_ (kN)	*N*_u,FE_ (kN)	*N*_u,exp_/*N*_u,FE_	*N*_u,exp_/*N*_u_	*N*_u,FE_/*N*_u_
AC-6-1	[38]	219 × 3.50 × 600	0.5	69.21	313	3055.00	3224.51	0.947	0.938	0.990
AC-6-2	[38]	219 × 3.50 × 600	0.5	69.21	313	3093.00	3224.51	0.959	0.950	0.990
AC-6-3	[38]	219 × 3.50 × 600	0.5	69.21	313	3073.00	3224.51	0.953	0.944	0.990
CFST3-0-1-8-a	[39]	138 × 2.72 × 420	1	40.52	299.4	1104.00	1047.60	1.054	1.093	1.037
CFST3-0-1-8-b	[39]	138 × 2.69 × 420	1	40.52	299.4	1115.00	1040.98	1.071	1.111	1.037
CFST3-0-1-8-c	[39]	138 × 2.69 × 420	1	40.52	299.4	1095.00	1040.98	1.052	1.091	1.037
C-S-R	[40]	165 × 3.54 × 480	0.5	50.00	368	1739.94	1909.81	0.911	0.925	1.015
C1	[41]	508 × 8.76 × 1500	1	72.36	355.8	19,662.20	18,617.9	1.056	0.998	0.945
C1	[42]	127 × 4.20 × 390	0.3	72.06	298.3	1552.50	1448.78	1.072	1.051	0.981
C2	[42]	127 × 4.20 × 390	0.6	70.38	298.3	1568.00	1468.97	1.067	1.070	1.002
C3	[42]	127 × 4.20 × 390	1	63.89	298.3	1462.50	1441.01	1.015	1.040	1.025
CA-1	[43]	88 × 2.50 × 285	0.1	29.14	342.7	517.53	547.57	0.945	1.031	1.091
CA-2	[43]	88 × 2.50 × 285	0.2	28.20	342.7	509.70	545.92	0.934	1.024	1.097
CA-3	[43]	88 × 2.50 × 285	0.3	32.36	342.7	522.24	544.09	0.960	1.011	1.053
CA-4	[43]	88 × 2.50 × 285	0.4	33.82	342.7	521.73	542.07	0.962	0.997	1.036
CA-5	[43]	88 × 2.50 × 285	0.5	31.51	342.7	519.93	539.67	0.963	1.013	1.052
CA-6	[43]	88 × 2.50 × 285	0.6	30.33	342.7	517.23	537.34	0.963	1.018	1.058
CA-7	[43]	88 × 2.50 × 285	0.7	35.86	342.7	530.88	534.81	0.993	0.995	1.002
CA-8	[43]	88 × 2.50 × 285	0.8	36.96	342.7	533.13	531.96	1.002	0.989	0.987
CA-9	[43]	88 × 2.50 × 285	0.9	34.30	342.7	538.08	529.05	1.017	1.021	1.004
CA-10	[43]	88 × 2.50 × 285	1	38.43	342.7	540.96	525.88	1.029	0.990	0.962
CB-1	[43]	112 × 2.00 × 360	0.1	29.14	357.2	639.63	669.04	0.956	1.054	1.102
CB-2	[43]	112 × 2.00 × 360	0.2	28.2	357.2	670.41	667.08	1.005	1.117	1.111
CB-3	[43]	112 × 2.00 × 360	0.3	32.36	357.2	677.64	664.94	1.019	1.072	1.052
CB-4	[43]	112 × 2.00 × 360	0.4	33.82	357.2	676.59	662.59	1.021	1.051	1.029
CB-5	[43]	112 × 2.00 × 360	0.5	31.51	357.2	673.65	659.92	1.021	1.076	1.054
CB-6	[43]	112 × 2.00 × 360	0.6	30.33	357.2	629.16	657.43	0.957	1.019	1.064
CB-7	[43]	112 × 2.00 × 360	0.7	35.86	357.2	660.00	655.00	1.008	0.999	0.991
CB-8	[43]	112 × 2.00 × 360	0.8	36.96	357.2	662.73	652.16	1.016	0.989	0.973
CB-9	[43]	112 × 2.00 × 360	0.9	34.30	357.2	660.09	649.79	1.016	1.016	1.000
CB-10	[43]	112 × 2.00 × 360	1	38.43	357.2	679.68	646.94	1.051	0.996	0.948
C1-1	[44]	114 × 1.80 × 400	0.25	40.08	300.0	656.00	655.18	1.001	1.078	1.077
C2-1	[44]	114 × 1.80 × 400	0.5	40.11	300.0	688.00	647.42	1.063	1.128	1.062
C3-1	[44]	114 × 1.80 × 400	0.75	38.86	300.0	640.00	632.67	1.012	1.065	1.053
C4-1	[44]	114 × 1.70 × 400	1	35.02	300.0	557.00	587.28	0.948	1.007	1.061
cfst8-30-0.5-1	[45]	140 × 2.71 × 420	0.5	44.4	309.0	1131.00	1153.39	0.981	1.035	1.056
cfst8-30-0.5-2	[45]	140 × 2.79 × 420	0.5	44.4	309.0	1139.00	1186.13	0.960	1.027	1.069
cfst8-30-0.5-3	[45]	140 × 2.83 × 420	0.5	44.4	309.0	1070.00	1180.90	0.906	0.957	1.057
cfst8-30-1-1	[45]	140 × 2.80 × 420	1	41.1	309.0	1102.00	1136.38	0.970	1.026	1.058
cfst8-30-1-2	[45]	140 × 2.73 × 420	1	41.1	309.0	1098.00	1121.57	0.979	1.036	1.059
cfst8-30-1-3	[45]	140 × 2.64 × 420	1	41.1	309.0	1118.00	1091.54	1.024	1.074	1.049
cfst8-50-1-1	[45]	140 × 2.72 × 420	1	65.0	309.0	1447.00	1524.47	0.949	1.056	1.113
cfst8-50-1-2	[45]	140 × 2.69 × 420	1	65.0	309.0	1398.00	1528.15	0.915	1.025	1.120
cfst8-50-1-3	[45]	140 × 2.81 × 420	1	65.0	309.0	1421.00	1546.02	0.919	1.024	1.114
cfst12-30-0.5-1	[45]	140 × 3.85 × 420	0.5	44.4	335.3	1365.00	1477.94	0.924	0.974	1.055
cfst12-30-0.5-2	[45]	140 × 3.86 × 420	0.5	44.4	335.3	1453.00	1481.02	0.981	1.035	1.055
cfst12-30-0.5-3	[45]	140 × 3.81 × 420	0.5	44.4	335.3	1351.00	1465.91	0.922	0.970	1.053
cfst12-30-1-1	[45]	140 × 3.96 × 420	1	41.1	335.3	1414.00	1493.01	0.947	1.018	1.074
cfst12-30-1-2	[45]	140 × 3.85 × 420	1	41.1	335.3	1437.00	1459.93	0.984	1.053	1.070
cfst12-30-1-3	[45]	140 × 3.84 × 420	1	41.1	335.3	1433.00	1456.81	0.984	1.052	1.069
cfst12-50-1-1	[45]	140 × 3.78 × 420	1	65.0	335.3	1550.00	1777.14	0.872	0.938	1.076
cfst12-50-1-2	[45]	140 × 3.92 × 420	1	65.0	335.3	1725.00	1816.21	0.950	1.026	1.080
cfst12-50-1-3	[45]	140 × 3.88 × 420	1	65.0	335.3	1749.00	1806.76	0.968	1.045	1.080
cfst15-30-0.5-1	[45]	133 × 4.57 × 400	0.5	44.4	302.0	1385.00	1471.08	0.941	1.024	1.088
cfst15-30-0.5-2	[45]	133 × 4.61 × 400	0.5	44.4	302.0	1377.00	1481.14	0.930	1.013	1.090
cfst15-30-0.5-3	[45]	133 × 4.66 × 400	0.5	44.4	302.0	1387.00	1493.71	0.929	1.013	1.091
cfst15-30-1-1	[45]	133 × 4.56 × 400	1	41.1	302.0	1386.00	1453.04	0.954	1.051	1.102
cfst15-30-1-2	[45]	133 × 4.61 × 400	1	41.1	302.0	1387.00	1466.15	0.946	1.045	1.104
cfst15-30-1-3	[45]	133 × 4.62 × 400	1	41.1	302.0	1357.00	1467.43	0.925	1.021	1.104
C2-1	[46]	114 × 1.84 × 396	0.5	43.9	300.0	636.50	688.42	0.925	0.982	1.062
C4-2	[46]	114 × 1.70 × 401	1	35.9	300.0	557.50	594.73	0.937	0.994	1.061
Mean								0.978	1.024	1.049
COV.								0.048	0.042	0.041

**Table 3 materials-15-00154-t003:** Material and geometric properties of specimens for FE parametric study.

*D* (mm)	*L* (mm)	*t* (mm)	*r*	*f*_cu,0_ (MPa)	*f*_s_ (MPa)
500	1500	3	0.00, 0.25, 0.50, 0.75, 1.00	30, 50	235
500	1500	3	0.00, 0.25, 0.50, 0.75, 1.00	50, 70	345
500	1500	3	0.00, 0.25, 0.50, 0.75, 1.00	70	420
500	1500	6	0.00, 0.25, 0.50, 0.75, 1.00	30, 50	235
500	1500	6	0.00, 0.25, 0.50, 0.75, 1.00	50, 70	345
500	1500	6	0.00, 0.25, 0.50, 0.75, 1.00	70	420
500	1500	10	0.00, 0.25, 0.50, 0.75, 1.00	30, 50	235
500	1500	10	0.00, 0.25, 0.50, 0.75, 1.00	50, 70	345
500	1500	10	0.00, 0.25, 0.50, 0.75, 1.00	70	420

**Table 4 materials-15-00154-t004:** Range of applicability of design equations for circular CFST/RACST columns.

Code	Yield Strength of Steel (MPa)	Compressive Strength of Concrete (MPa)	Diameter to Thickness Ratio
ACI-318-11 [50]	All	All	$\leq 8\sqrt{E_{s}/f_{y}}$
T/CECS 625-2019 [51]	235 ≤ *f*_y_ ≤ 460	30 ≤ *f*_cu_*^r^*_,150_ ≤ 50	≤135(235/*f*_y_)
EC4 (2004) [52]	235 ≤ *f*_y_ ≤ 460	20 ≤ *f*_c,150_ ≤ 50 or 25 ≤ *f*_cu,150_ ≤ 60	≤90(235/*f*_y_)
AIJ (1997) [53]	235 ≤ *f*_y_ ≤ 355	*f*_c,100_ ≤ 58.8	≤1.5(23500/*F*)
AISC 360-16 (2016) [54]	*f*_y_ ≤ 525	21 ≤ $f_{c}^{\prime }$ ≤ 69	≤0.31(*E*_s_/*f*_y_)

**Table 5 materials-15-00154-t005:** Summary of available formulae in well-known international codes.

Reference	Formulae	Remarks	Total	Average Values (*N*_u,exp_/*N*_u,ref_)	Dispersion Coefficient
ACI-318-11 (2011) [50]	$N_{u,\mathrm{ACI}}=A_{s}f_{s}+0.85f_{c}^{\prime }A_{c}$	Circular	61	1.429	0.060
T/CECS 625-2019 [51]	$\begin{array}{l} N_{u,\text{T/CECS625-2019}}=f_{sc,r}(A_{c}+A_{s})\\ f_{\mathrm{sc},r}=(1.14+1.02\xi_{r,o})f_{c,r}\\ \xi_{r,o}=\frac{A_{s}f}{A_{c}f_{c,r}} \end{array}$	Circular	44	1.188	0.046
EC4 (2004) [52]	$\begin{array}{l} N_{u,\mathrm{EC}4}=\eta_{a}A_{s}f_{y}+A_{c}f_{c}^{\prime }(1+\eta_{c}\frac{t}{D}\frac{f_{y}}{f_{c}^{\prime }})\\ \eta_{a}=0.25(3+2\bar{\lambda })\leq 1.0;\\ \eta_{c}=4.9-18.5\bar{\lambda }+17{\bar{\lambda }}^{2}\geq 1.0;\\ \bar{\lambda }=\sqrt{\frac{N_{pl,Rk}}{N_{cr}}};N_{pl,Rk}=A_{s}f_{y}+A_{c}f_{c}^{\prime }\\ N_{cr}=\frac{\pi^{2}(E_{s}I_{s}+0.6E_{c}I_{c})}{L^{2}} \end{array}$	Circular	48	1.103	0.037
AIJ (1997) [53]	$\begin{array}{l} N_{u,\mathrm{AIJ}}=\gamma_{c}f_{c}^{\prime }A_{c}+(1+\eta )f_{y}A_{s}\\ \eta =0.27,\gamma_{c}=0.85 \end{array}$	Circular	48	1.267	0.037

**Table 6 materials-15-00154-t006:** Summary of available formulae in other scholars’ literature.

Reference	Formulae	Remarks	Total	Average Values (*N*_u,exp_/*N*_u,ref_)	Dispersion Coefficient
Chen, Z. P. [16]	$\begin{array}{l} N_{u}=\eta f_{\mathrm{ck}}A_{c}(1+0.99\sqrt{\xi }+1.11\xi )\\ \xi =\frac{A_{s}f}{A_{c}f_{\mathrm{ck}}},\eta =0.901+0.084r-0.154r^{2} \end{array}$	Circular	61	1.035	0.072
Chen, M.C. [23]	$\begin{array}{l} N_{u}=f_{\mathrm{scy}}^{r}(A_{c}+A_{s}),f_{scy}^{r}=f_{\mathrm{scy}}^{}/k\\ f_{scy}^{}=(1.14+1.02\xi )f_{ck},\xi =\frac{A_{s}f}{A_{c}f_{c}}\\ k=A{\left(e^{-\xi }\right)}^{2}+Be^{-\xi }+C\\ A=0.1059r+0.0066\\ B=0.12r+0.0021,C=0.054r+1.0044 \end{array}$	Circular	61	1.268	0.064
Xu, W. [24]	$\begin{array}{l} N_{u}=N_{\text{u,GB50936-2014}}/\theta \\ N_{\text{u,GB50936-2014}}=f_{\mathrm{sc}}(A_{s}+A_{c}),\theta =0.904+0.012r\\ f_{\mathrm{sc}}=(1.212+B\xi +C\xi^{2})f_{c},\;\xi =\frac{A_{s}f}{A_{c}f_{c}}\\ B=0.176f/213+0.974,C=-0.104f_{c}/14.4+0.031 \end{array}$	Circular	61	1.025	0.079

## Data Availability

No new data were created or analyzed in this study. Data sharing is not applicable to this article.

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
