# Peer review of "Confinement Effect and Efficiency of Concentrically Loaded RACFCST Stub Columns"

_materials, 2021, doi:10.3390/ma15010154_

Round 1

Reviewer 1 Report

This paper presents a comprehensive numerical study on the axial compressive behaviour of recycled aggregate concrete-filled circular steel tubular columns (RACFCST). The topic is very innovative and addresses an important issue in practice. The numerical model is well established and validated against test data. In-depth analysis is then presented to reveal the confinement effect within RACFCST columns, with the calculation methods proposed. Publication is recommended. This reviewer only has minor suggestions for the improvement of this manuscript,

  1. Line 28 of Introduction, change ‘The past several decades has…’ to ‘The past several decades have…’?
  2. The caption of Table 4, it might be better to change ‘national codes’ to ‘international codes’.
  3. In Fig. 2, when illustrating the boundary conditions, it is suggested to use consistent wording for ‘the top end’ and ‘the bottom’, i.e., ‘the bottom end’ is suggested.
  4. In Fig. 5, two ‘FE curves’ were shown in the legend, it is suggested to distinguish them using different labels.
  5. In Fig. 6(b), please use capital letter for ‘Steel strength’. In Fig. 12(b), please use lowercase letter for ‘ratio’.
  6. There are recent studies on the axial behaviour of RACFST with circular sections where the effects of confinement factor are discussed and partial factors in strength design are proposed. E.g., Axial compressive behaviour and design calculations on recycled aggregate concrete-filled steel tubular (RAC-FST) stub columns, ES. It would be great if this can be reflected in the current study as well.
  7. In Fig. 10, it might not be appropriate to use Mises stress for the concrete since it is not a metal material.
  8. The authors stated in Section 2.1 that the value of friction coefficient for the tangential behavior in the bond-slip action between steel tube and infilled RAC was set as 0.5. How was this value determined?
  9. As stated in the last paragraph of Section 2.2, plastic damage model in ABAQUS was adopted. The concrete damaged plasticity (CDP) model is proposed for and usually used in fresh concrete, please clarify how they combined with the constitutive equations (Eqs. 1 and 2) proposed by Ding as well as how they were used in the modelling. In addition, please discuss how the relationships of compressive or tensile damage parameters with inelastic strains were defined in the software Abaqus for the damage evolution of CDP, especially for the differences between fresh concrete and RAC.

Reviewer 2 Report

The paper is well-structured and easy to follow.

The content of the paper is solid and outstanding.

This paper presents technical information about an analytical (finite element) and experimental study of the structural performance of RACFCST stub columns under axial load. As part of the results documented in this manuscript, a simple and practical equation was reported to calculate the ultimate bearing capacity of RACFCST stub columns. In general, the paper is well-written, and the structure is proper. The paper may contribute to the Materials Journal. However, the Authors must perform the following revisions.

  1. The first sentence of the Abstract section is a little bit confusing. In this sense, please change the sentence “Recycled aggregate concrete-filled steel tubular (RACFST) columns is widely recognized as an efficient structural member” to “Recycled aggregate concrete-filled steel tubular (RACFST) columns are widely recognized as efficient structural members”.
  2. In the whole manuscript, the word “behaviours” must be replaced by “behaviour”. Please remember that must be declared in singular the mechanical behaviour.
  3. By the end of the Abstract section, please try to be more specific on the main contribution of this paper to the Materials Journal.
  4. In the Introduction section, the first sentence of the third paragraph must be changed from “One of the most promising structural members in its form can provide sufficient confinements on concrete and reduce their shrinkage and creep is the concrete-filled steel tubular (CFST) columns [15]” to “One of the most promising structural members in its form that can provide sufficient confinements on concrete and reduce their shrinkage and creep is the concrete-filled steel tubular (CFST) columns [15]”.
  5. In the Introduction section, the second sentence of the third paragraph must be changed from “Correspondingly, the recycled aggregate concrete-filled steel tabular (RACFST) columns are a convenient way can suppress the defects of RAC and improve their mechanical performance” to “Correspondingly, the recycled aggregate concrete-filled steel tabular (RACFST) columns are a convenient way that can suppress the defects of RAC and improve their mechanical performance”.
  6. By the end of the Introduction section, particularly in the last paragraph of such a section, please increase the discussion about the main contribution of this paper to the Materials Journal.
  7. In Section 2.1, please justify the use of the 8-node reduced integral format 3D solid element (C3D8R) to model the steel tube, infilled RAC and the loading plate. Why did you select such a finite element?
  8. Why did you use ABAQUS instead of other type of FE software? Please justify this in Section 2.1.
  9. Equation (2) is composed of so many other Equations, please divide every of the equations reported in Equation (2). In addition, why are those equations inside a curly brace? Please revise this.
  10. In the Conclusions section, please include the limitations of the study presented in this paper.
